# Genetic diversity of local and introduced cassava germplasm in Burundi using DArTseq molecular analyses

Niyonzima Pierre[1,2]*, Lydia Nanjala Wamalwa[1], William Maina Muiru[1], Bigirimana Simon[2], Edward Kanju[3], Morag Elizabeth Ferguson[4], Malu Muia Ndavi[5], Silver Tumwegamire[6]

**1** Department of Plant Science and Crop Protection, University of Nairobi, Nairobi, Kenya, **2** Institut des Sciences Agronomiques du Burundi (ISABU), Bujumbura, Burundi, **3** International Institute of Tropical Agriculture (IITA), Kampala, Uganda, **4** International Institute of Tropical Agriculture (IITA), Nairobi, Kenya, **5** International Fund for Agriculture Development (IFAD), Rome, Italy, **6** International Institute of Tropical Agriculture (IITA), Kigali, Rwanda

* pierreniyonzima@ymail.com, pierreniyonzima2014@gmail.com

**Data Availability Statement:** All relevant data are within the manuscript and its Supporting Information files.

## Abstract

In Burundi most small-scale farmers still grow traditional cassava landraces that are adapted to local conditions and have been selected for consumer preferred attributes. They tend to be susceptible, in varying degrees, to devastating cassava viral diseases such as Cassava Brown Streak Disease (CBSD) and Cassava Mosaic Disease (CMD) with annual production losses of US$1 billion. For long term resistance to the disease, several breeding strategies have been proposed. A sound basis for a breeding program is to understand the genetic diversity of both landraces and elite introduced breeding cultivars. This will also assist in efforts to conserve landraces ahead of the broad distribution of improved varieties which have the possibility of replacing landraces. Our study aimed at determining the genetic diversity and relationships within and between local landraces and introduced elite germplasm using morphological and single nucleotide polymorphism (SNP) markers. A total of 118 cultivars were characterized for morphological trait variation based on leaf, stem and root traits, and genetic variation using SNP markers. Results of morphological characterization based on Ward's Method revealed three main clusters and five accessions sharing similar characteristics. Molecular characterization identified over 18,000 SNPs and six main clusters and three pairs of duplicates which should be pooled together as one cultivar to avoid redundancy. Results of population genetic analysis showed low genetic distance between populations and between local landraces and elite germplasm. Accessions that shared similar morphological traits were divergent at the molecular level indicating that clustering using morphological traits was inconsistent. Despite the variabilities found within the collection, it was observed that cassava germplasm in Burundi have a narrow genetic base.

**Funding:** The funders had no role in study design, data collection and analysis, decision to publish, or preparation of the manuscript.

## Introduction

Cassava was the most important staple crop in Burundi in 2019, with production of 2.41 million tons followed by fruits, bananas, sweet potatoes and vegetables [1]. It is grown mainly by small scale farmers throughout low, medium and high-altitude areas for human consumption. The root crop is eaten in the form of "imikembe", "ubuswage" and processed into flour for 'ugali' while leaves are used as vegetables or sauce [2]. Production of cassava doubled between 2010 to 2013 [1] but since then, there has been a steady reduction in production, mainly due to cassava brown streak disease (CBSD) and cassava mosaic disease (CMD). This is exacerbated by a lack of improved resistant cassava cultivars and the continued use of local susceptible landraces. The need to determine the genetic composition of local landraces and enhance the frequency of resistance genes within the local gene pool is a priority.

Breeding approaches for clonally propagated crops include variety introduction, germplasm assembly and maintenance, clonal selection and hybridization [3]. Breeding methods of cassava are defined by its genetic variability, the mode of reproduction and the breeding objectives. Cassava is a highly heterozygous species and presents substantial segregation in the first generation progenies, that are then evaluated through phenotypic mass selection [4]. The methods developed for self-pollinating crops are applicable to cassava with some modifications because of its specific characteristics. There is no classic genetic improvement methods initiated for vegetative propagated crops [5]. The main genetic improvement methods used in cassava are the assembly of the germplasm and selection followed by hybridization among selected elite clones [3, 6]. The introduction of varieties and selection are the most important breeding methods used in most of African countries [7]. However, crossing followed by selection of superior genotypes in the segregating population is the most universal method employed in cassava genetic improvement.

However, agricultural genetic diversity is imperative to provide a robust food security system able to adapt to pest, disease and environmental stresses [8] as well as to make genetic gains in plant breeding. It allows breeders to develop superior cultivars adapted to changing climatic conditions to meet end user demands. Understanding genetic diversity of species is the basis for a breeding program and to develop strategies for germplasm collection, management, conservation and improvement for food security and sustainable agricultural development [9–11]. Genetic diversity studies have been done for cassava using both morphological and molecular methods in other countries such as Brazil [12], Chad [13], Benin Republic [14], Nigeria [15], Tanzania [16] etc. but not Burundi.

Morphological markers are widely used to characterize cassava germplasm but the number of variants is limited, and this type of marker is influenced by the environment unlike molecular markers. Morphological markers have been used by Agre et al. [14] to study the genetic diversity and relationships among elite cassava cultivars in Benin and highlighted significant diversity and the most discriminating morphological parameters within the germplasm.

Initial work with molecular markers focused on SSR markers, RFLP markers, AFLP markers and RAPD markers. De Souza [17] reported the identification of Simple Sequence Repeat (SSR) and Amplified Fragment Length Polymorphisms (AFLP) markers linked to the CMD-resistance gene in cassava landraces and Random Amplified Polymorphic DNAs (RAPD) markers linked to resistance to anthracnose. SSR markers were developed and utilized to construct the genetic linkage map of cassava [18] and evaluate the genetic diversity of cassava [17, 19, 20]. The first genetic linkage map of cassava was constructed from F1 intra-specific cross using SSR, Restriction Fragment Length Polymorphisms (RFLP) and RAPD [18]. The frequency and number of alleles per SSR marker in the Puerto Rican cassava collection were determined [21]. RFLP, AFLP and RAPD markers were used to analyze the genetic diversity of

cassava [22–24]. Furthermore, study on the genetic diversity and relationships within cassava germplasm using SNPs markers, was done by Karim et al. [25]. The utilization of SNPs has gained popularity in recent years due to their abundance, ubiquitous nature, polymorphism and amenability to automation [26]. In cassava, SNPs have been used for genetic linkage mapping [27, 28], genome-wide association studies [15] and genetic diversity assessments [29]. Molecular markers associated with agronomic traits have contributed significantly to marker assisted cassava breeding programs [16, 30]. Diversity Arrays Technology (DArT) SNP markers, generated using DArTSeq technology for cassava were developed and reported as a tool for genotyping large germplasm collections [31] but this has not been used on Burundian cassava genotypes. DArT performs well in polyploid species and does not require any existing DNA-sequence information and can be used with little resources required for SNP platforms [32]. Thus, it is a sequence-independent genotyping method designed to detect genetic variation at several hundred genomic loci in parallel without relying on sequence information.

Our study sought to characterize and determine relationships of cassava local landraces and introduced elite genotypes, as well as identifying any putative duplicates, using morphological and molecular markers.

## Materials and methods

### Germplasm collection and establishment

One hundred local landraces of cassava were collected from four agro-ecological zones in Burundi, namely Imbo plain, Mumirwa slopes, east and north depressions, and Central plateau (Table 1), selected on the basis of their importance for growing cassava. The identification of cassava landraces was done in farmers' fields jointly by farmers and investigators based on a short discussion. The landraces were recorded under the name as given by the farmers. Once collected, the landraces were planted in field gene banks at two sites (Bukemba and Murongwe ISABU research stations) representing two major cassava growing regions in Burundi for morphological and molecular characterization. Bukemba is located at 03°59′ 54″ S and 30°4′49″ E, 1180.9 m.a.s.l. in Rutana province in southeastern of Burundi while Murongwe located at 03°11′36″S and 29° 53′47″E, 1523 m.a.s.l. in Gitega province in central of Burundi. Eighteen elite cassava genotypes earlier introduced to Burundi (Table 2) were also planted at the same sites. Single row plots with five plants spaced one meter between and within rows were used for each genotype in the trial. No fertilizer or irrigation were provided and weeds were managed throughout the growing period.

### Morphological characterization of cassava local landraces and introduced elite germplasm

Seventeen qualitative agro-morphological traits were evaluated (Table 3) using cassava descriptors described by Fukuda *et al.* [33] at 3, 6, 9 and 12 months after planting (MAP). Color and pubescence on apical leaves were recorded earlier rather than later to avoid obscured traits due to damage by cassava green mites that normally infest cassava at later stages of plant growth. At 6 MAP, data on the shape of central leaf lobe and color of the leaf and petiole, and petiole orientation were recorded by taking a leaf from the mid-height stem position. At 9 MAP, data on prominence of foliar scars, color of stem cortex and color of stem exterior were recorded from the middle third of the plant. Color of stem cortex was visualized by shallow cut and peel back of the epidermis as described by Fukuda *et al.* [33]. Distance between leaf scars was measured from the middle part of stem on the middle third of the plant, where the scars are not flat. Measurement was made along the stem and the distance was divided by the number of nodes in the measured section to obtain the mean internode length. Data on the stem's growth habit was recorded either

**Table 1. Cassava landraces and their region of origin within Burundi.**

| Name of accession | Agro-ecological zone | Name of accession | Agro-ecological zone | Name of accession | Agro-ecological zone |
|---|---|---|---|---|---|
| Nakarasi ya congo | 1 | Gatarina | 3 | Mpamba | 4 |
| Nakarasi y'ikirundi | 1 | Serereka | 3 | Mabare | 4 |
| Gitamisi_1 | 1 | Bugiga annonciate_1 | 3 | Imiduga_1 | 4 |
| Muzinda | 1 | Yongwe_2 | 3 | Tabika | 4 |
| Kwezikumwe | 1 | Gitikatika | 3 | Yongwe ederi | 4 |
| Rumonge | 1 | Gifunzo caritsa_1 | 3 | Umukurajoro | 4 |
| Mbubute | 1 | Gifunzo caritsa_2 | 3 | Rukokora | 4 |
| Yagata | 1 | Fyiroko | 3 | Kinazi dorothee1 | 4 |
| Niga | 1 | Munebwe | 3 | Gasu | 4 |
| Ibigororoka | 1 | Ndoha | 3 | Inagitembe | 4 |
| Maguruyinkware_1 | 1 | Maguruyinkware_2 | 3 | Umutuburano | 4 |
| Mwarabu | 1 | Rumarampunu | 3 | Gitamisi_2 | 4 |
| Rushishwa | 1 | Imikabika | 3 | Rubona_2 | 4 |
| Sosomasi | 1 | Hanyesi | 3 | Nakarasi_1 | 4 |
| Myezisita | 1 | Rubona_1 | 3 | Surupiya | 4 |
| Zegura | 1 | Bwome devote1 | 3 | Sogota | 4 |
| Igipila | 1 | Umuyobera | 4 | Nabuseri | 4 |
| Igikoshi | 1 | Gasahira | 4 | Imirundi | 4 |
| Nakarasi_2 | 1 | Mbwayasaze | 4 | Imizariya | 4 |
| Solange | 2 | Kidihe_1 | 4 | Maguruyinkware_3 | 4 |
| Yongwe_1 | 2 | Bunwa | 4 | Umutakabumba | 4 |
| Kibembe_1 | 2 | Inarubono | 4 | Mugerera Yvonne_1 | 4 |
| Criolina | 2 | Ntunduguru | 4 | Mugerera Yvonne_2 | 4 |
| Matara | 2 | Kigoma | 4 | Kidihe_2 | 4 |
| Sisiriya | 2 | Imijumbura | 4 | Nyawera | 4 |
| Ruvuna | 2 | Nyabisindu anastasie_1 | 4 | Nyamugari sophie_1 | 4 |
| Butoke | 2 | Kabumbe | 4 | Mukecuru | 4 |
| Kiganda | 2 | Gasasa | 4 | Fundiko | 4 |
| Ntabahungu | 2 | Yongwe_3 | 4 | Umuhendangurube | 4 |
| Kibembe_2 | 2 | Mutsindekwiburi | 4 | Sagarara | 4 |
| Munengera | 3 | Murozi | 4 | Imiduga_2 | 4 |
| Mwotsi_2 | 3 | Umusimbaruzi | 4 | Mwotsi_1 | 4 |
| Berita | 3 | Bukarasi | 4 | Kavyiro | 4 |
| Ntegagakoko | 3 | - | - | - | - |

1 = Imbo plain, 2 = Mumirwa slopes, 3 = East and north depressions, 4 = Central plateau.

as straight or zig-zag, and color of the end branches of the adult plant was observed on the top 20 cm of the plant. At 12 MAP, observations on color of root cortex, color of root-pulp, external color of storage root and root taste were taken. Root cortex color and color of root-pulp were visualized by removing the skin of the root and by transversal cutting of the root.

## Molecular characterization of cassava local landraces and introduced elite germplasm

In terms of DNA extraction; four disks of approximately 5mm diameter were collected from young fresh leaf samples. These were dried in an oven overnight at 45˚C and shipped to

**Table 2. Introduced elite germplasm in Burundi and their country of origin.**

| Variety name | Country of origin |
|---|---|
| KBH2002/066 | Tanzania |
| Pwani | Tanzania |
| Mkumba | Tanzania |
| KBH2006/026 | Tanzania |
| Kizimbani | Tanzania |
| Kiroba | Tanzania |
| Albert | Tanzania |
| Okhumelela | Mozambique |
| Orera | Mozambique |
| Eyope | Mozambique |
| Tajirika | Kenya |
| F10-30-R2 | Kenya |
| Kibandameno | Kenya |
| TZ 130 | Uganda |
| Nase14 | Uganda |
| Nase1 | Uganda |
| Nase3 | Uganda |
| MM96/5280 | Burundi |

Intertek in Australia for DNA extraction, before being forwarded to Diversity Array Technologies for genotyping using DArTseq. DNA quality and quantity were checked on a 0.8% agarose gel. Libraries were constructed at Diversity Arrays Technology in Canberra, Australia according to DArTseq™ complexity reduction method through digestion of genomic DNA and ligation of barcoded adapters [34]. DArT uses a genotyping by sequencing DArTseq™ technology, providing rapid, high quality and affordable genome profiling, even from the most complex polyploid genomes [34, 35]. SNP marker scoring was achieved using DArTsoft14 which is an in-house marker scoring pipeline based on algorithms [34]. Two types of DArTseq: SilicoDArT and SNP markers were both scored as binary markers for presence or absence (1 and 0 respectively).

## Data analyses

Morphological data (S3 Table) was analyzed using the Statistical Package for the Social Sciences (SPSS) software (IBM SPSS Statistics for Windows version 20.0, IBM Corp, Armonk, NY.). Dissimilarity matrix was used to determine the relationship among accessions. The structure of morphological variation was visualized using ascending hierarchical clustering (AHC) based on data and Ward's Method to plot a dendrogram [25, 13]. Morphological traits (S3 Table) distribution was determined using Microsoft (MS) Excel (2016). Generated SNP data (S1 Table) were cleaned in MS Excel by removing all genotypes with >5% missing data and monomorphic SNPs. Hamming's single distance between genotypes (S2 Table) was calculated using KDCompute, Version 1.5.2 beta and hierarchical clustering done by Ward's method for dendrogram (https://www.rdocumentation.org/packages/dartR). Generated SNP data (S1 Table) were imported into DartR and then filtered for repeatability, monomorphic loci, call rate per locus, single locus per sequence tag and call rate per individual [36]. To better identify putative duplicated genotypes and to determine cut-off, known duplicate cassava genotypes were included with the samples genotyped. To assess the population statistics, the

**Table 3. Qualitative traits used to characterize 118 cassava genotypes.**

| Trait observed | Trait acronym | Sore code | Data entry |
|---|---|---|---|
| Color of apical leaves | CAL | 3 = light green; 5 = dark green; 7 = purplish green; 9 = purple | 3 MAP |
| Pubescence on apical leaves | PAL | 0 = absent, 1 = present | 3 MAP |
| Shape of central leaflet | SCL | 1 = ovoid; 2 = elliptical-lanceolate; 3 = obovate-lanceolate; 4 = oblong-lanceolate; 5 = lanceolate; 6 = linear; 7 = pandurate; 8 = linear-pyramidal; 9 = linear-pandurate; 10 = linear-hostatilobalate | 6 MAP |
| Petiole color | PC | 1 = yellowish-green, 2 = green, 3 = reddish-green, 5 = greenish-red, 7 = red, 9 = purple | 6 MAP |
| Leaf color | LC | 3 = light green; 5 = dark green; 7 = purple green; 9 = purple | 6 MAP |
| Petiole orientation | PO | 1 = inclined upwards, 3 = horizontal, 5 = inclined downwards, 7 = irregular | 6 MAP |
| Prominence of foliar scars | PFS | 3 = semi-prominent, 5 = prominent | 9 MAP |
| Color of stem cortex | CSC | 1 = orange, 2 = light green, 3 = dark green | 9 MAP |
| Color of stem epidermis | CSEp | 1 = cream, 2 = light brown, 3 = dark brown, 4 = orange | 9 MAP |
| Color of stem exterior | CSEx | 3 = orange, 4 = green-yellowish, 5 = golden, 6 = light brown, 7 = silver, 8 = gray, 9 = dark brown | 9 MAP |
| Distance between leaf cars | DBLS | 3 = short ($\leq$ 8 cm), 5 = medium (8–15 cm), 7 = long ($\geq$15 cm) | 9 MAP |
| Growth habit of stem | GHS | 1 = Straight, 2 = Zig-zag | 9 MAP |
| Color of end branches of adult plant | CEBAP | 3 = Green, 5 = Green-purple, 7 = Purple | 9 MAP |
| Color of root cortex | CRC | 1 = White or cream, 2 = Yellow, 3 = Pink, 4 = Purple | 12 MAP |
| Color of root-pulp | CRP | 1 = white; 2 = cream; 3 = yellow; 4 = orange; 5 = pink | 12 MAP |
| External color of storage root | ECSR | 1 = white or cream; 2 = yellow; 3 = light brown; 4 = dark brown | 12 MAP |
| Root taste | RT | 1 = Sweet, 2 = Intermediate, 3 = Bitter | 12 MAP |

MAP = Months after planting.

observed heterozygosity (Ho) was calculated using mean 'hobs' function and expected heterozygosity (He) using Hs function in the R package 'Adegenet' [37–39]. The pair wise fixation index (Fst) among populations was calculated using StAMPP package in R and the output value indicated existence or not of differentiation between populations where <15% indicate

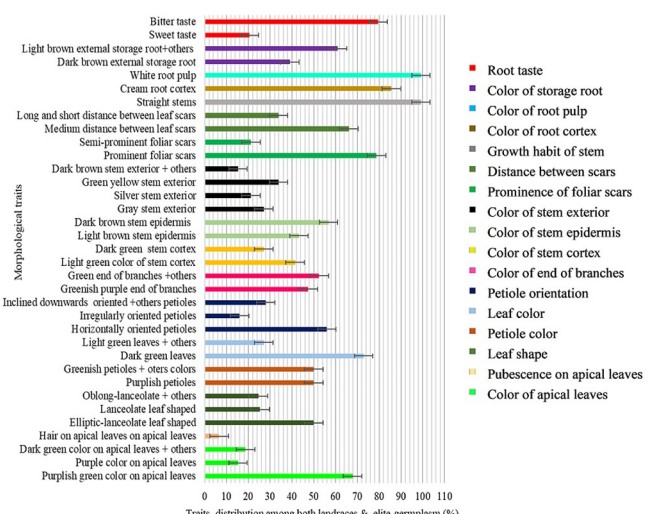

**Fig 1. Morphological traits distribution among both landraces and elite germplasm with error bars indicating whether differences are statistically significant.**

low differentiation, 0.15<Fst<0.25 indicate moderate differentiation and >25% indicate high differentiation [40]. Genetic relationships of landrace and introduced cassava genotypes were assessed by estimation of hamming distance between genotypes using dartR in KDCompute as described by Hoque et al. [41] and the population structure was assessed using Discriminant Analysis of Principal Components (DAPC) method. Single distance matrix (S2 Table) was exported as a csv file and imported into DARwin v6.0.21 [42] to construct a dendrogram to estimate the genetic relationships.

## Results

### Morphological traits of cassava local landraces and improved elite germplasm

**Leaf traits.** There was a diversity of color on apical leaves for the cassava genotypes. Most accessions (68%) had purplish green color as the dominant color for apical leaf (Fig 1) mostly dominated by landraces (64.4%) (Fig 2), and few of the elite germplasm (3.4%) (Fig 3). About 19% and 15% of the accessions had dark-green and purple apical leaf color, respectively. Less than 7% of the accessions had pubescence on apical leaves (Fig 1). The shape of the leaves also had variations where 50% of the accessions had elliptic-lanceolate as the dominant shape (Fig 1) and mostly among the landraces (44.1%) (Fig 2). Obovate lanceolate, pandurate, lanceolate-pandurate and linear-pyramidal leaf shapes were rare, and altogether observed in 6.8% of the accessions (Fig 2). The color of petioles varied among the accessions, where purple color was the most frequent (50%). Most landraces had purple color (44.9%) compared to only 5.1% of the elite germplasm (Figs 2 and 3) respectively. Other petiole colors were observed, including yellowish-green, green, green purple, purple yellow, red-green, and red (Fig 1). Dark green color (72.9%) (Fig 1) was the dominant leaf color observed in most accessions of which 68.6% were landraces (Fig 2). The most frequent petiole observed was horizontally oriented (56%) (Fig 1) and more so for landraces (52.5%) (Fig 2). Color of the end branches of adult plants was mostly greenish purple among accessions (47.5%) (Fig 1), which was the most frequent color among the landraces (39%) (Fig 2). However, green and purple colors were also observed among the accessions (Fig 1).

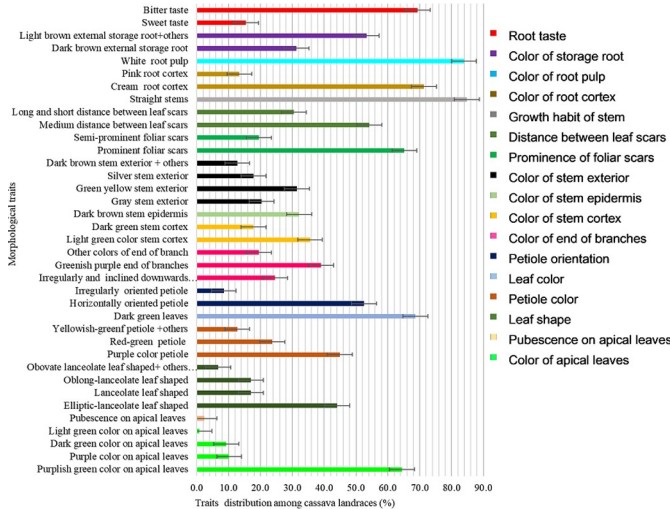

**Fig 2. Morphological traits distribution among the cassava landraces with error bars indicating whether differences are statistically significant.**

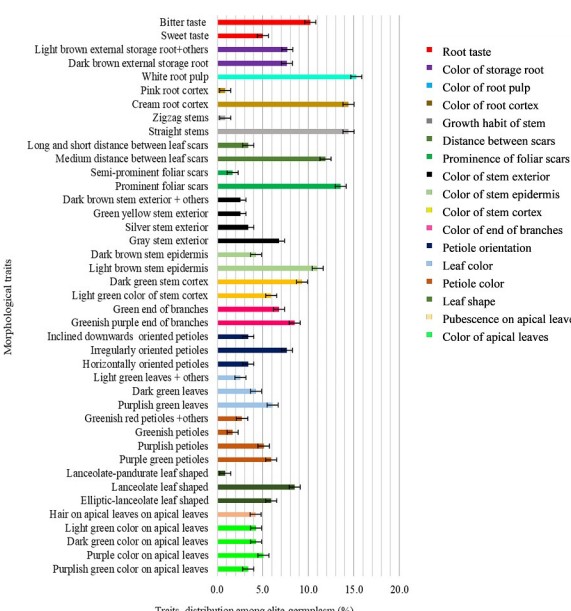

**Fig 3. Morphological traits distribution among the elite germplasm with error bars indicating whether differences are statistically significant.**

**Stem traits.** Most accessions (41.5%) had light green stem cortex color (Fig 1), mostly dominated by landraces (35.6%) (Fig 2) and a few (5.9%) by the elite germplasm (Fig 3). Dark green stem color was found on 27.1% of the 118 accessions (Fig 1). Epidermis color was diverse, where more than 56% accessions were dark brown (Fig 1) mostly landraces (32.2%) (Fig 2). The rest of the accessions (43%) had light brown stem epidermis. Color of stem exterior was mostly green yellow with 33.9% of the accessions (Fig 1) mostly landraces (Fig 2), followed by grey (27.1%) (Fig 1), also dominated by landraces (Fig 2). Silver color of stem exterior then followed at 21.2% (Fig 1) mostly landraces (Fig 2). Other stem exterior colors recorded were dark brown, orange and green colors (Fig 1).

Foliar scars were prominent among 78.8% of the accessions while 21.2% had semi prominent foliar scars. Accessions with prominent foliar scars were mostly landraces (65.2%) (Fig 2) while the elite germplasm with prominent foliar scars comprised only 13.6%. The distance between leaf scars varied within the cassava accessions where 66% had medium distance (8–15 cm) (Fig 1) comprised mostly by landraces (Fig 2), 34% had long ($\geq$ 15 cm) and short ($\leq$ 8 cm) distance. All the accessions had straight stem growth habit except one (Orera) that had the zigzag stem habit (Fig 1).

**Root traits.** Cream root cortex color was recorded among 85.7% of the accessions (Fig 1) and almost all the elite germplasm (14.4%) belonged to this group (Fig 3). Pink root cortex has been observed on some accessions (Fig 1). All the accessions had white pulp color except Solange that had yellowish root pulp (Figs 1–3). Dark brownish external storage root color was the most frequent (39.0%) among the accessions (Fig 1) mostly the landraces (31.4%) (Fig 2). Bitter taste was noted for 79.5% of the accessions (Fig 1), mainly among both the landraces (Fig 2) and the elite germplasm (Fig 3).

## Hierarchical clustering of cassava local landraces and improved elite germplasm

Ascending hierarchical clustering analysis based on morphological traits and Ward's method showed three major clusters (I, II and III) (Fig 4) following the horizontal line at a dissimilarity

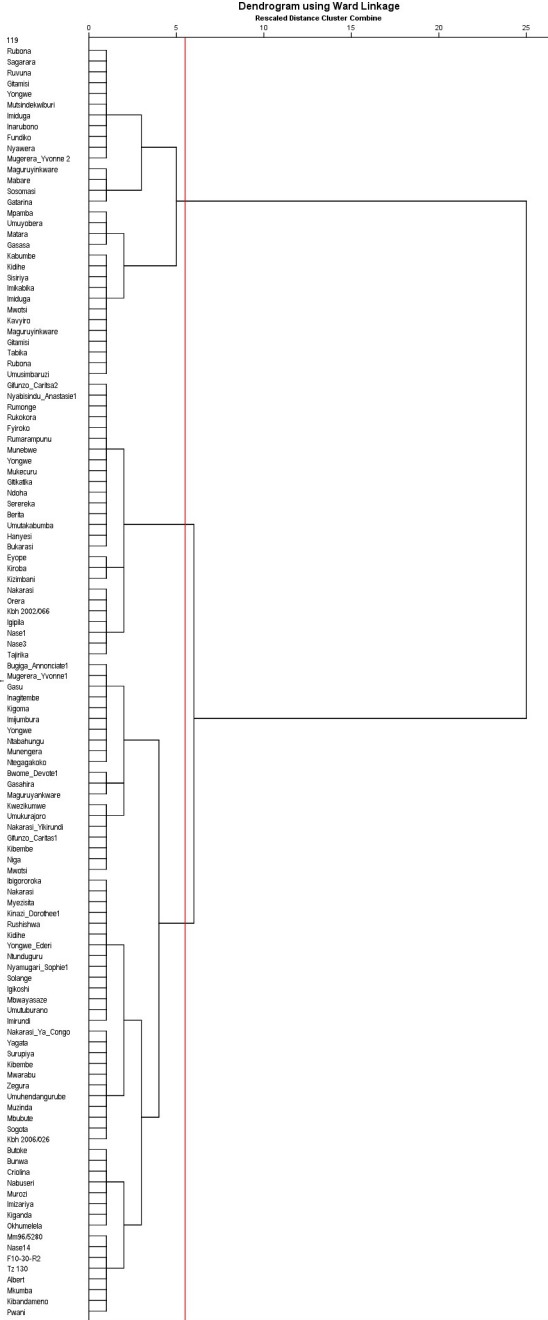

**Fig 4. Phenotypic classification of cassava accessions based on the Ward's method at a dissimilarity level of 6.**

level of 6. Cluster I containing 31 accessions (all local landraces) had two sub-clusters. Cluster II consisted of 26 accessions (3 sub-clusters) and was composed of local landraces and elite genotypes. Sub-cluster III of cluster II consisted of five elite genotypes; Tajirika, Nase 1, Nase 3, KBH2002/066 and Orera, and two local landraces (Nakarasi and Igipila), sub-cluster II of 3 elite genotypes namely Kizimbani, Kiroba and Eyope while sub-cluster I consisted of 16 local landraces (Fig 4). Cluster III was the largest with two sub-clusters consisting of 51 local landraces and ten elite genotypes. Elite genotypes under this category were KBH2006/026,

Okhumelela, MM96/5280, Nase14, F10-30-R2, TZ130, Albert, Mkumba, Kibandameno and Pwani (Fig 4).

## Genetic relationship among cassava genotypes using DArT analyses

Results from DartR analysis showed 72 unique genotypes, 39 genotypes presented similar SNP profile (Fig 5) following the cut off (green line) calculated from the distance matrix based on an average value of known duplicates. Putative duplicates accessions were grouped in 16 classes, each of them with different clones (Fig 5). Genotypic classification of accessions based on Ward's method showed six major clusters (Fig 5) at dissimilarity level of 1.0 (red line). Cluster I had two elite genotypes, Pwani and Mkumba alongside five known duplicate checks: Pwani_2, Pwani_3_SB101, Mkumba_1, Pwani_1, Mkumba_2_SB102 (Fig 5). Cluster II had nine genotypes consisting of four local (Nakarasi ya congo, Rumonge, Munembwe and Gitamisi) and five elite genotypes, namely KBH2006/026, Tajirika, KBH2002/066, Kizimbani and Okhumelela (Fig 5). Cluster II had five duplicates namely, Tajirika-2, KBH 2002/026/1, KBH 2002/026/2, Tajirika-5CP-Kephis and KBH 2002-066-SB103 (Fig 5). Cluster III and V consisted of eight and seven accessions, respectively, all local landraces. Cluster IV was composed of 58 accessions sub clustering into two mains groups that were sub clustered in different subgroups showing many similarities (Fig 5). Cluster IV consisted of 50 local landraces and eight elite genotypes including Orera, F10-30-R2, Kibandameno, Albert, Okhumelela, MM96/5280, Nase 14 and TZ130 (Fig 5). Cluster VI consisted of 33 local landraces. Paired similar accessions that fell into this category were Igikoshi and Munengera, Sosomasi and Igipila, Mwotsi, Mwarabu and Mwzisita, Bunwa and Kigoma, Maguruyinkware-2 and Rumaramuntu, Ndoha and Imikabika, and Bugiga annociate 1 and Gifunzo caritas 2 (Fig 5). Discriminant Analysis of Principal Components (DAPC) showed five subpopulations with the third subpopulation distinguished against the remaining four others (Fig 6). As for the structure, the different inferred populations do not match to the agro-ecological zones of sampling; each population consisted of individuals from at least two or three agro-ecological zones. The fifth subpopulation was the largest with 51 genotypes, followed by the first subpopulation consisting of 32 genotypes and sub population two with 20 genotypes (Fig 6). The third and fourth are the lowest subpopulation consisting respectively of 6 and 8 genotypes (Fig 6).

## Assessment of the population statistics of the genotypes

Assessment was done within and between populations to determine existence of any relationships. The output values of calculated pair wise fixation index (Fst) among all populations were <15% indicating low differences between populations (Table 4 and Fig 7). Similarly, the Principal coordinate analysis (PCoA) showed little genetic variation among the five

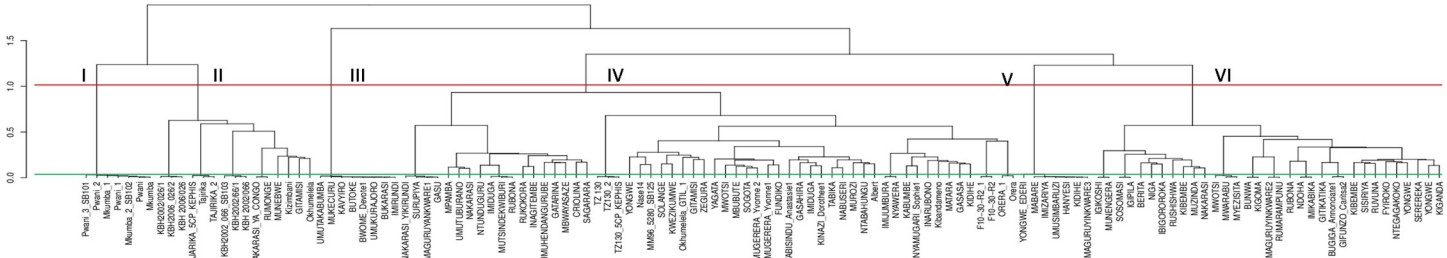

**Fig 5. Genotypic classification of accessions based on the Ward's method at dissimilarity level of 1.0 (red line), the green line determining the threshold for putative and known duplicates.**

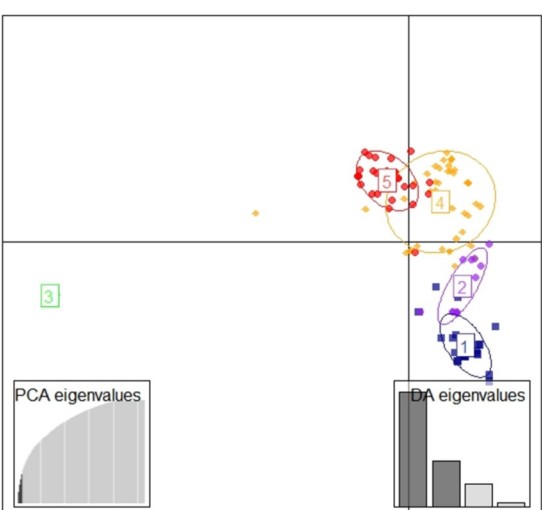

**Fig 6. Population structure according to the DAPC using 18,124 SNPs.** The first two components are displayed graphically (each sub-population is differentiated by color).

populations, the first two axes explained 17.2% of the total variation, corresponding to 9.8 and 7.4% for the first and second axis, respectively (Fig 8). Results showed pair wise fixation index of 0.071, 0.095, 0.073 and 0.083 between elite genotypes and local landraces of Imbo plain, Mumirwa slopes, North east depressions and Central plateau, respectively, that showed little variation (Table 4). Between local landraces of Imbo plain and Mumirwa slopes, North East (NE) Depressions and Central plateau, the pair wise fixation index was 0.010, 0.023 and 0.020, respectively, indicating very low differentiation between populations. Pair wise fixation index between landraces of Mumirwa Slopes and NE Depressions, between landraces of Central plateau and NE Depressions were 0.027 and 0.028 respectively (Table 4).

Within population, output values for pair wise fixation index were greater than 25% for all populations indicating high differentiation between genotypes (Table 5). Pair wise fixation index of 0.59, 0.60, 0.57, 0.59 and 0.56 was noted within elite genotypes and landraces of Imbo plain, Mumirwa slopes, NE depressions and Central plateau, respectively, indicating high variation between genotypes (Table 5). The heterozygosity was calculated per marker and population (S8 Table), where observed heterozygosity (Ho) was greater than expected heterozygosity (He) in all populations except elite genotypes, indicating a suspected mixing of previously isolated populations (Table 5).

**Table 4. Pairwise fixation index between landraces from different locations.**

|  | Elite genotypes | Landraces of Imbo Plain | Landraces of Mumirwa Slopes | Landraces of NE Depressions | Landraces of Central Plateau |
|---|---|---|---|---|---|
| Elite genotypes | - |  |  |  |  |
| Landraces of Imbo Plain | 0.071 | - |  |  |  |
| Landraces of Mumirwa Slopes | 0.095 | 0.010 | - |  |  |
| Landraces of NE Depressions | 0.073 | 0.023 | 0.027 | - |  |
| Landraces of Central Plateau | 0.083 | 0.020 | 0.001 | 0.028 | - |

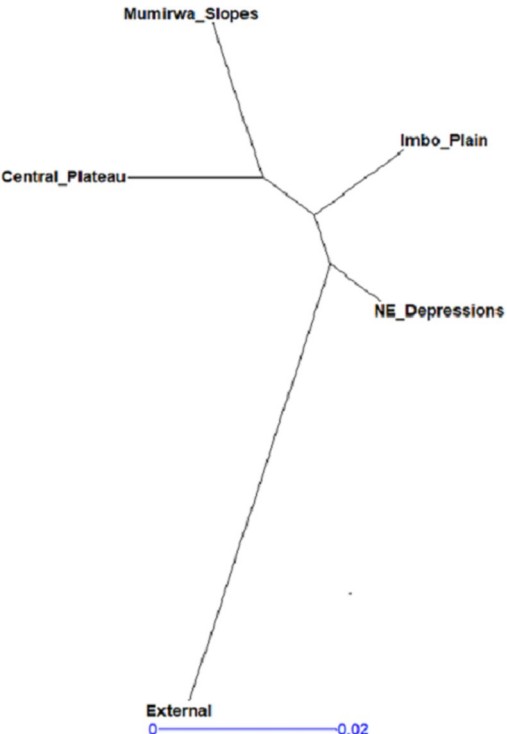

**Fig 7. Genetic relationships between cassava populations based on Nei's genetic distance.**

## Comparison of results from morphological and molecular dendrograms

Morphological classification clustered accessions into three main groups, whereas molecular analysis clustered accessions into six groups. All genotypes in clusters I and II for morphological classification method and clusters III, V and VI for genetic classification method were local landraces. Cluster I in the genetic classification method only consisted of elite genotypes

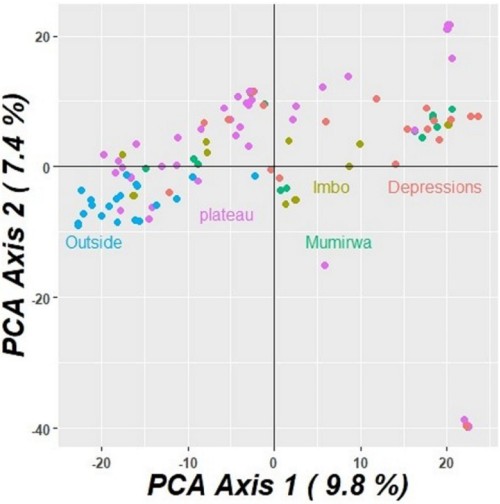

**Fig 8. PCoA of cassava populations based on 18,124 SNP's.**

**Table 5. Fixation index and heterozygosity within population.**

| Population | Fixation Index F within population | Observed heterozygosity (Ho) | Expected heterozygosity (He) |
|---|---|---|---|
| Elite genotypes | 0.59 | 0.25 | 0.27 |
| Landraces of Imbo Plain | 0.60 | 0.27 | 0.25 |
| Landraces of Mumirwa slopes | 0.57 | 0.27 | 0.25 |
| Landraces of NE Depressions | 0.59 | 0.26 | 0.25 |
| Landraces of Central Plateau | 0.56 | 0.26 | 0.25 |

(Pwani and Mkumba) while clusters II and IV for the same method and cluster III for the morphological clustering method contained both local landraces and elite genotypes.

## Discussion

### Morphological traits

Morphological characterization based on leaf traits (color of apical leaves, color of end branches of adult plant, pubescence on apical leaves, shape of central leaf lobe, petiole color, prominence of foliar scars, distance between leaf scars, leaf color, petiole orientation), stem traits (color of stem cortex, color of stem epidermis, color of stem exterior, growth habit of stem) and root traits (color of root cortex, color of root-pulp, external color of storage root and root taste) were diverse among the cassava landraces as well as the elite germplasm studied. These traits are very interesting and can be used in breeding and in identifying varieties.

**Leaf traits.** The leaf traits play an important role in cultivar identification and are more relevant for selection of cassava varieties suitable for the leafy vegetable markets. Leaf shape is one such an important trait as it affects leaf area and hence light interception which can directly affect root yield [43]. Asare et al. [44] and Agre et al. [14] reported respectively that leaf shape and color were the most important variables to distinguish cassava accessions and that farmers identify their cassava cultivars based on the traits related to leaf and stem color. The analysis revealed that apical leaves of 68% of cultivars were colored purplish green as dominant color and the mature leaves of 72.9% of cultivars are colored dark green as dominant color. Phosaengsri et al. [45] reported that leaf color plays an important role in predicting fresh root weight as nearly 90% of the dry matter (or biomass) of a plant is produced by leaves. Study of Khumaida et al. [46] revealed that dark green leaf color would increase the weight of tubers per plant, thus, can be useful in predicting root yield estimates of several cassava genotypes. Analysis also revealed few accessions colored light green on apical leaves, having hairs and with central leaflet shaped linear- pyramidal, which were comparable to those obtained by Nadjiam et al. [13]. According to Ehleringer et al. [47], presence of hair on apical leaves reduces leaf light absorbance, heat load, and consequently lower leaf temperatures and transpiration rates. On the other hand, presence of hairs on leaves lowers photosynthetic activity, and therefore lowers the yield. In addition, most of the end branches of adult plants of most of the genotypes were colored greenish purple. Comparable results were obtained by Eze et al. [48] who reported the predominant greenish purple color of end branches in adult plants in cassava varieties from Nigeria.

**Stem traits.** Most of the landrace accessions had stem epidermis and stem exterior colored light brown and grey, respectively. Similar findings were reported by Kosh-Komba et al. [49] who studied the diversity of cassava in Central Africa Republic and underlined clusters of accessions characterized by stems colored light brown and gray. Our results revealed that more than 78% of characterized cassava genotypes had prominent foliar scars while more than

21% had semi-prominent foliar scars. According to Adu et al. [50] and Banoc et al. [51], when cuttings are planted in soil, roots developed from the foliar scars as well as lateral branches. Furthermore, the distance between leaf scars determines the number of scars per unit stem length and indeed the number of lateral branches.

**Root traits.** Presence of accessions showing cream color on root cortex and cream color on root pulp confirmed the presence of genotypes with low levels of beta-carotene, the precursor of Vitamin A [52]. Furthermore, yellow cassava roots are associated with high levels of proteins in leaves and, therefore, improving cassava for high beta-carotene content could also improve the overall nutritional value of the crop [52]. Almost 80% of the accessions were found to have a bitter taste, suggesting that roots from these cultivars may have high levels of toxic cyanogenic glucosides and therefore, must be processed prior to consumption as suggested by Chiwona-Karltun et al. [53]. Furthermore, the presence of different colors of the external storage roots can be used to differentiate cassava accessions.

## Molecular characterization

The analysis based on molecular characterization clustered accessions into six main clusters indicating varying genetic distances between genotypes. Clusters I contained two elite genotypes, Pwani and Mkumba that shared all genetic characteristics, suggestive of putative duplicates. The putative duplicates clones detected within cluster I such as Pwani and Mkumba, which were distributed by the "New Cassava varieties and Clean Seed to Combat CMD and CBSD" (5CP) project have previously been shown to be the same genotype by Ferguson et al. [16].

Clusters III, V and VI had eight, seven and 33 of all landraces, respectively, suggesting that in each group landraces had more similar genetic traits while the elite genotypes and landraces shared less similar genetic traits. Under cluster II, five elite genotypes namely Okhumelela, Kizimbani, KBH2002/066, Tajirika and KBH2006/026 clustered together with four local landraces namely Gitamisi, Munembwe, Rumonge and Nakarasi while in cluster IV, seven elite genotypes namely F-10-30-R2, Kibandameno, Orera, Albert, MM96/5280, Nase14 and TZ130 clustered together with 51 local landraces, suggesting that some local landraces and elite genotypes have similar genetic characteristics. Thus, the local landraces that belonged to the two clusters (II and IV) could be possible sources of resistance or they could be related due to genetic elements that control other traits other than resistance. Furthermore, in cluster II, a pair of accessions namely: Nakarasi ya congo and Rumonge shared all genetic characteristics, indicating that these accessions can be assumed to be putative duplicates clones.

## Assessment of the population statistics

Population statistics analysis within and between populations that determine existence of any relationships revealed little variation between populations, and between introduced genotypes and local landraces of Imbo plain, Mumirwa slopes, North east depressions and Central plateau, suggesting exchange of germplasm between farmers in these regions. Much of the variation was accounted for within individual which is typical of a highly heterozygous out crossing species.

## Morphological and molecular analysis

Accessions that shared similar morphological characteristics were distinct at the molecular level, indicating that the resolution provided by morphological traits is lower than with molecular markers. These results are in agreement with findings of Sujii et al. [54] and Feldberg et al. [55] who reported that plants showing similar morphological characteristics could be very distinct at molecular level. In addition, Sujii et al. [54] and Darkwa et al. [56] reported that

clustering using morphological traits is less reliable due to the influence of the environment and plant growth stage on their expression and the limited number of markers to distinguish entities. This phenomenon could explain the changing and clustering observed in comparing the hierarchical cluster dendrograms for the morphological and molecular traits [56]. Accessions Kizimbani, Rumonge, Munembwe and Tajirika were grouped in clusters II for both morphological and molecular characterization. Pwani and Mkumba were grouped in cluster I for molecular characterization while cluster III in morphological characterization together with Kibandameno, Albert, TZ130, F10-30-R2, Nase 14 and MM96/5280 in one of the sub clusters, demonstrating the varied discriminative powers of the two methods of characterization. The putative duplicates clones detected within clusters shows that some genotypes such as Pwani and Mkumba, belonging to elite germplasm and Imiduga, Mutsindekwiburi and Rubona belonging to local landraces are several copies, thus could be pooled together as one cultivar. Difference of number of clusters between the two methods of characterization was due to the number of specific traits and genetic variations for phenotypic and genotypic classification respectively. However, the phenotypic classification dealt with a small number of traits, seventeen qualitative agro-morphological traits while the genotypic classification dealt with more than 18 000 SNPs, hence the genotypic classification showed a lot of differences between accessions. Comparable results have been reported by Albuquerque et al. [57] and Arnaud-Haond et al. [58] while identifying duplicate accessions based on multi-locus analysis, and concluded that accessions presenting similar SNP profiles were assumed to be putative duplicates as each multi locus genotype corresponded to a single genotype.

## Conclusion

The aim of this study was to determine the morphological and genotypic polymorphism in the local landraces and characterize elite cassava genotypes as well as identifying duplicates. Morphological and molecular characterization showed distinct classes of cultivars and within each class, sub classes with similar SNP profiles were identified. Accessions having very close similar characteristics namely Pwani and Mkumba, and Imiduga, Mutsindekwiburi and Rubona should be considered as putative duplicates, hence, need to be pooled together as one cultivar. Despite the variabilities found within the collection, it was concluded that cassava landraces in Burundi as well as the introduced clones present a narrow genetic base.

## Supporting information

**S1 Table. SNP data from DArtR Seq. 0: Homozygous reference allele, 1: Homozygous alternate allele, 2: Heterozygous alleles, -: Missing value, M1 to M18124: Numbers of SNP's.**
(CSV)

**S2 Table. Hamming distance matrix.**
(CSV)

**S3 Table. Morphological traits.**
(XLSX)

**S4 Table. Data on morphological traits distribution among both landraces and elite germplasm, used to build graph.**
(CSV)

**S5 Table. Data on morphological traits distribution among the cassava landraces, used to build graph.**
(CSV)

**S6 Table. Data on morphological traits distribution among the elite germplasm, used to build graph.**
(CSV)

**S7 Table. Data used to plot dendrogram based on morphological traits and Ward's method.**
(CSV)

**S8 Table. Heterozygosity_per_marker.**
(CSV)

## Acknowledgments

The cassava research team from ISABU is acknowledged for their support in trial management and data collection. Ms Chepkoech, Jackline, a laboratory technician at the Integrated Genotyping Service and Support (IGSS) of the Bioscience Eastern and Central Africa supported the molecular analysis work.

## Author Contributions

**Conceptualization:** Niyonzima Pierre, Lydia Nanjala Wamalwa, William Maina Muiru, Silver Tumwegamire.

**Data curation:** Niyonzima Pierre, Morag Elizabeth Ferguson.

**Formal analysis:** Niyonzima Pierre, Morag Elizabeth Ferguson.

**Funding acquisition:** Bigirimana Simon, Malu Muia Ndavi, Silver Tumwegamire.

**Investigation:** Niyonzima Pierre, Lydia Nanjala Wamalwa, William Maina Muiru, Silver Tumwegamire.

**Methodology:** Niyonzima Pierre, Lydia Nanjala Wamalwa, William Maina Muiru, Edward Kanju, Silver Tumwegamire.

**Project administration:** Bigirimana Simon, Malu Muia Ndavi, Silver Tumwegamire.

**Supervision:** Lydia Nanjala Wamalwa, William Maina Muiru, Silver Tumwegamire.

**Validation:** Lydia Nanjala Wamalwa, William Maina Muiru, Silver Tumwegamire.

**Visualization:** Niyonzima Pierre, Lydia Nanjala Wamalwa, William Maina Muiru, Silver Tumwegamire.

**Writing – original draft:** Niyonzima Pierre, Lydia Nanjala Wamalwa, William Maina Muiru, Edward Kanju, Morag Elizabeth Ferguson, Silver Tumwegamire.

**Writing – review & editing:** Niyonzima Pierre, Lydia Nanjala Wamalwa, William Maina Muiru, Bigirimana Simon, Edward Kanju, Morag Elizabeth Ferguson, Malu Muia Ndavi, Silver Tumwegamire.

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
