## [Decision Letter · Decision Letter 0]

31 Aug 2021

PONE-D-21-23112

Genetic diversity of local and introduced cassava germplasm in Burundi using DArTseq molecular analyses

PLOS ONE

Dear Dr. pierre,

Thank you for submitting your manuscript to PLOS ONE. After careful consideration, we feel that it has merit but does not fully meet PLOS ONE’s publication criteria as it currently stands. Therefore, we invite you to submit a revised version of the manuscript that addresses the points raised during the review process.

We look forward to receiving your revised manuscript.

Kind regards,

Amit Kumar Singh

Academic Editor

PLOS ONE

Journal Requirements:

2. Please upload a new copy of figures 4 & 5 as the detail is not clear. Please follow the link for more information: https://blogs.plos.org/plos/2019/06/looking-good-tips-for-creating-your-plos-figures-graphics/" https://blogs.plos.org/plos/2019/06/looking-good-tips-for-creating-your-plos-figures-graphics/

3. In your Methods section, please provide additional location information, including geographic coordinates for the data set if available.

4. In your Methods section, please provide additional information regarding the permits you obtained for the work. Please ensure you have included the full name of the authority that approved the field site access and, if no permits were required, a brief statement explaining why.

[This work was sponsored by the International Institute of Tropical Agriculture (IITA) under the ‘CBSD (Cassava Brown Streak Disease) Control Project’. The project is funded by the International Fund for Agricultural Development (IFAD) and the “Institut des Sciences Agronomiques du Burundi” (ISABU) is the main partner in Burundi. The cassava research team from ISABU are acknowledged for their support in trial management and data collection. Ms Chepkoech, Jackline, a laboratory technician at the Integrated Genotyping Service and Support (IGSS) of the Bioscience Eastern and Central Africa supported the molecular analysis work.]

 [The funders had no role in study design, data collection and analysis, decision to publish, or preparation of the manuscript.]

Additional Editor Comments:

Reviewers have found your manuscript interesting for consideration in PLOS ONE, however they have made some suggestions which need to be incorporated before it can be accepted for publication.

Reviewers' comments:

Reviewer's Responses to Questions

**Comments to the Author**

1. Is the manuscript technically sound, and do the data support the conclusions?

Reviewer #1: Yes

Reviewer #2: Yes

2. Has the statistical analysis been performed appropriately and rigorously? 

Reviewer #1: Yes

Reviewer #2: Yes

3. Have the authors made all data underlying the findings in their manuscript fully available?

Reviewer #1: Yes

Reviewer #2: Yes

4. Is the manuscript presented in an intelligible fashion and written in standard English?

Reviewer #1: Yes

Reviewer #2: Yes

5. Review Comments to the Author

Reviewer #1: The research paper presents an analysis of the diversity in cassava landraces in Barundi. Overall, the manuscript is well written, and provides insight into the morphological and molecular diversity present in the cassava germplasm of Barundi. The experiments are well planned and executed and after analysis, some important conclusions have been arrived upon which can be utilized for designing further strategies for improvement of this plant species. However, some suggestions are enlisted below:

1. In the abstract the authors mention that the study is aimed at developing a core set of cassava germplasm from the local varieties (Line 30), however no mention of this objective is made in the manuscript text, nor has any analysis been done to this end. The manuscript deals with the genetic diversity present in the landraces and elite lines. Appropriate corrections may be incorporated in the abstract.

2. Discussion on molecular characterization and duplicate identification may elaborated.

3. Overall, the discussion section may be improved.

4. Cassava plants were planted in two locations. Did the authors observe any morphological variation among the same plants at the two locations? If yes, authors could indicate the effect of the environment/ soil etc. on the morphological characteristics.

5. Authors have studied cassava landraces from four different sites along with elite introduced germplasm. Can they carry out a population structure analysis for the four different populations and present it graphically? The analysis may bring out important information on the extent of mixing that may have happened in these genotypes over the course of time.

6. Authors may also support/depict the genetic diversity and genetic classifications they have observed using Principal Component Analysis (PCA).

7. Sentence framing and language may be checked in some parts of the introduction and discussion.

Reviewer #2: The MS “Genetic diversity of local and introduced cassava germplasm in Burundi using DArTseq molecular analyses” presents the results of original research and experiment for morphological and DArT marker analyses and statistics have been performed for assessment of cassava genetic resources in Burundi in sufficient detail.

ALthough, authors may consider the following comment for improvement of the manuscript. Diversity Arrays Technology is a high-throughput genotyping method for preferential sampling of gene-rich regions that is achieved through the use of methylation sensitive restriction enzymes. The authors must use the DArT markers to perform redundancy analysis and assemble the non-redundant sequences for homology. This will provide DArT sequences incorporation into publicly genomics resources. Attribution of putative gene functions can also be done for the non-redundant DArT marker sequences that may aid in identifying markers with putative disease resistance-related functions esp. Cassava Brown Steak Disease (CBSD) and Cassava Mosaic Disease (CMD). This will assist in employing the available data in an enhanced way.

6. PLOS authors have the option to publish the peer review history of their article (what does this mean?). If published, this will include your full peer review and any attached files.

Reviewer #1: No

Reviewer #2: **Yes: **Manjusha Verma

---

## [Author Response · Author response to Decision Letter 0]

6 Dec 2021

Responses to the Reviewers: Genetic diversity of local and introduced cassava germplasm in Burundi using DArTseq molecular analyses

Reviewer #1: 

1. In the abstract the authors mention that the study is aimed at developing a core set of cassava germplasm from the local varieties (Line 30), however no mention of this objective is made in the manuscript text, nor has any analysis been done to this end. The manuscript deals with the genetic diversity present in the landraces and elite lines. Appropriate corrections may be incorporated in the abstract. 

Response: We agree with the reviewer and accordingly any mention of ‘core’ has been removed from the abstract. 

2. Discussion on molecular characterization and duplicate identification may elaborated. 

Response: In response to this, results of the discriminant analysis of principal components and principal coordinate analysis have been included. 

3. Overall, the discussion section may be improved. 

Response: A paragraph on the assessment of population statistics has been included.

4. Cassava plants were planted in two locations. Did the authors observe any morphological variation among the same plants at the two locations? If yes, authors could indicate the effect of the environment/ soil etc. on the morphological characteristics. 

Response: No morphological variation among the same genotypes at the two locations has been observed.

5. Authors have studied cassava landraces from four different sites along with elite introduced germplasm. Can they carry out a population structure analysis for the four different populations and present it graphically? The analysis may bring out important information on the extent of mixing that may have happened in these genotypes over the course of time.

Response: A discriminant analysis of principal components has been included and discussed. 

6. Authors may also support/depict the genetic diversity and genetic classifications they have observed using Principal Component Analysis (PCA).

Response: This has been included.

7. Sentence framing and language may be checked in some parts of the introduction and discussion.

Response: This has been addressed

Reviewer #2: 

Diversity Arrays Technology is a high-throughput genotyping method for preferential sampling of gene-rich regions that is achieved through the use of methylation sensitive restriction enzymes.

The authors must use the DArT markers to perform redundancy analysis and assemble the non-redundant sequences for homology. This will provide DArT sequences incorporation into publicly genomics resources. Attribution of putative gene functions can also be done for the non-redundant DArT marker sequences that may aid in identifying markers with putative disease resistance-related functions esp. Cassava Brown Steak Disease (CBSD) and Cassava Mosaic Disease (CMD). 

This will assist in employing the available data in an enhanced way.

Response: Thank you for the comment. All analysis of sequence data, alignments, and SNP calling was carried out by DArT. They performed redundancy analysis and assembled the non-redundant sequences for homology. There were 18,000 SNP markers, making it beyond the scope of this manuscript to assign gene functions. Your comment is appreciated.

---

## [Editor Report · Decision Letter 1]

17 Dec 2021

Genetic diversity of local and introduced cassava germplasm in Burundi using DArTseq molecular analyses

PONE-D-21-23112R1

Dear Dr. pierre,

We’re pleased to inform you that your manuscript has been judged scientifically suitable for publication and will be formally accepted for publication once it meets all outstanding technical requirements.

Kind regards,

Amit Kumar Singh

Academic Editor

PLOS ONE

---

## [Editor Report · Acceptance letter]

12 Jan 2022

PONE-D-21-23112R1 

Genetic diversity of local and introduced cassava germplasm in Burundi using DArTseq molecular analyses 

Dear Dr. pierre:

I'm pleased to inform you that your manuscript has been deemed suitable for publication in PLOS ONE. Congratulations! Your manuscript is now with our production department. 

Kind regards, 

on behalf of

Dr Amit Kumar Singh 

Academic Editor

PLOS ONE